# ART-FR: MASKED AUTO-REGRESSIVE TRANSFORMER FOR FACE RESTORATION

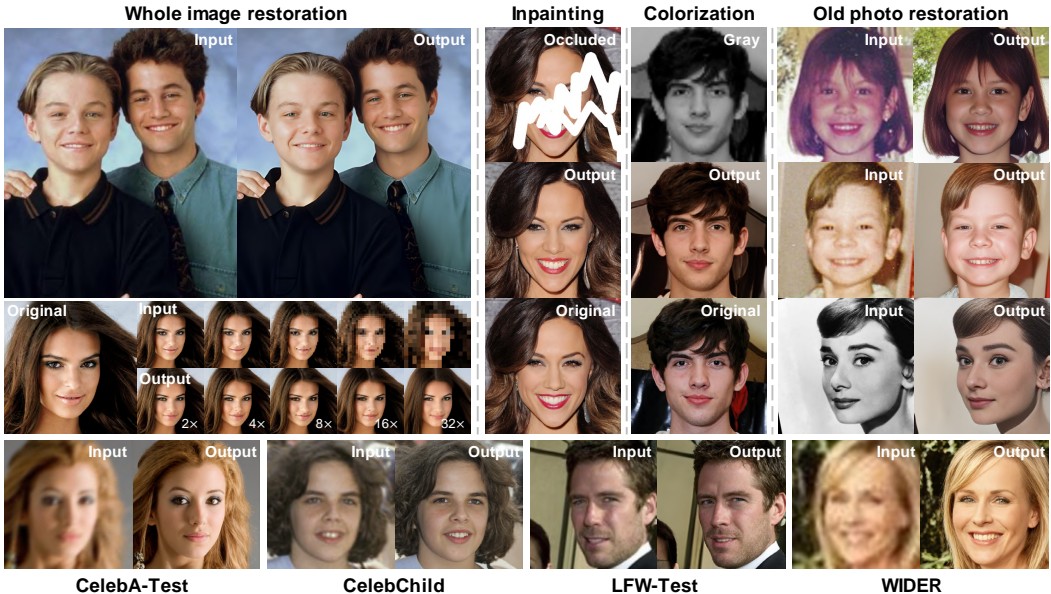

Figure 1: **ART-FR restoration results for various tasks.** ART-FR can achieve robust face restoration performance across various tasks and different levels of degradation, as well as different datasets.

## ABSTRACT

Restoring authentic facial features from low-quality images presents an extremely challenging task, due to the intricate real-world degradations and the inherently ill-posed nature of the problem. Existing methods, which utilize a codebook prior, help alleviate the complexity of the restoration process and produce visually plausible outcomes. However, these methods struggle to accurately capture the mapping between low-quality (LQ) and high-quality (HQ) images in the discrete latent space, leading to suboptimal results. Inspired by the success of auto-regressive generation paradigm in discrete modeling problems (*e.g.*, large language models), we propose an Auto-Regressive Transformer based Face Restoration (ART-FR) method to mitigate this mapping challenge. Specifically, with the aid of a visual tokenizer, we reformulate the face restoration task as a conditional generation problem within the discrete latent space. Furthermore, a masked generative image transformer is employed to model the distribution of this latent space, conditioned on LQ features. Face restoration is subsequently performed in the latent space through iterative sampling, with the HQ image reconstructed using a pretrained decoder. Extensive experimental validation demonstrates ART-FR exhibits superior performance across various benchmark datasets.

## 1 INTRODUCTION

Blind face restoration seeks to reconstruct high-quality (HQ) facial images from low-quality (LQ) inputs that have been degraded by intricate factors such as blur, noise, and compression. The inherently ill-posed nature of this problem poses substantial challenges in recovering missing facial details

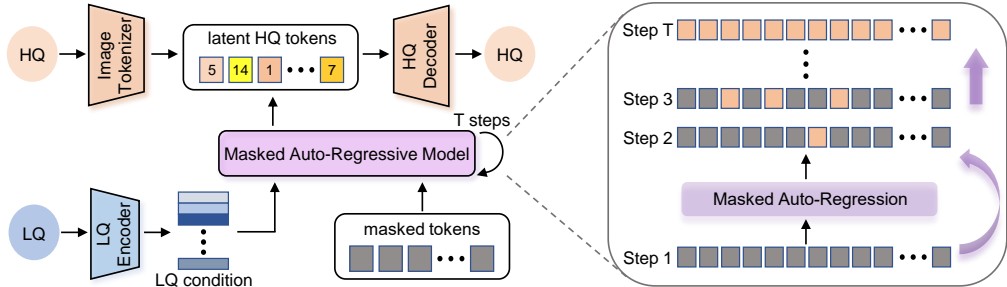

Figure 2: **An illustration of our motivation and proposed ART-FR framework.** Initially, an HQ image tokenizer is employed to generate a sequence of discrete tokens. Subsequently, a masked auto-regressive model is applied to estimate the distribution of these tokens, conditioned on LQ images. Through iterative predictions, the masked tokens are progressively transformed into corresponding HQ tokens that capture both semantic and textural information. Finally, an HQ decoder is utilized to reconstruct the high-quality image.

and generating photo-realistic textures. Despite the significant advancements in image restoration (IR) driven by sophisticated network architectures (Chen et al., 2023; Zamir et al., 2022; Li et al., 2023), end-to-end frameworks relying on minimizing fidelity measures (*e.g.*, pixel-wise losses) tend to produce over-smoothed results (Ledig et al., 2017). To address this issue, diverse generative priors have been introduced to preserve detailed facial information during the restoration process.

Recent advancements in this field have focused on leveraging generative models (Goodfellow et al., 2014; Ho et al., 2020), including Generative Adversarial Networks (GANs) and diffusion models, to enhance restoration quality. However, GAN-based approaches often encounter training difficulties and are prone to generating artifacts (Yang et al., 2021; Wang et al., 2021). Conversely, diffusion-based methods, while promising, introduce unwanted randomness as a consequence of incorporating random Gaussian noise (Wu et al., 2024; Sun et al., 2023). Furthermore, the computational expense is significant due to the necessity of executing numerous diffusion steps for the inference process.

With the aid of vector-quantized (VQ) autoencoder (Van Den Oord et al., 2017), state-of-the-art works have simplified the face restoration problem from a pixel-level task to code prediction task in discrete latent space, thereby reducing the complexity of the LQ-HQ mapping and demonstrating significant robustness against severe degradation. Nevertheless, existing methods still struggle to accurately model the mapping between LQ images and their corresponding discrete HQ latent representations. VQFR (Gu et al., 2022) employs a simple nearest-neighbor code matching technique, while Codeformer (Zhou et al., 2022) and DAEFR (Tsai et al., 2023) utilize an additional prediction network for the task. None of these methods accurately capture the discrete HQ latent representations due to the significant domain gap between the feature maps of LQ and HQ images.

Auto-regressive generation paradigm have recently spurred remarkable advancements in the modeling of discrete distributions. Notably, the GPT series (Radford et al., 2019; Brown, 2020) and other large language models (LLMs) have showcased exceptional comprehension and reasoning capabilities across diverse scenarios. Likewise, auto-regressive image synthesis methods have delivered high-quality image generation (Tian et al., 2024; Chang et al., 2022). Recognizing that the code-prediction problem of VQ-based face restoration aligns seamlessly with the Auto-Regressive Transformers' capability to model the distribution of discrete tokens, we reconceptualize face restoration as a generative task on discrete latent space, conditioned on LQ features. Under this framework, we propose our masked Auto-Regressive Transformer for Face Restoration, referred to as ART-FR.

The core concept of ART-FR is to establish a conditional auto-regressive modeling framework for restoration tasks, as depicted in Fig. 2. We adopt a two-stage learning strategy. First, following previous VQ-based methods, we train a visual tokenizer to discretize continuous image data into a grid of 2D tokens, which are subsequently flattened into a 1D sequence for Auto-Regressive (AR) learning. In the second stage, we integrate a bidirectional transformer for Masked Auto-regressive Modeling (Chang et al., 2022). Additionally, we treat LQ features (conditional information) as supplementary tokens to guide the auto-regressive generation process, mapping the masked tokens to HQ tokens. During inference, ART-FR generates predicted HQ tokens by iteratively sampling from the learned latent distribution, which are then decoded back into HQ images using the visual token decoder. Leveraging the generative capabilities of auto-regressive transformers, we can effectively mitigate the mapping challenge from LQ images to discrete HQ representations in latent space, also

maintaining an efficient restoration process. The main contributions of this work are summarized as follows:

- We reformulate the face restoration problem as a conditional generative task within a discrete latent space, facilitated by the use of a visual tokenizer. This approach significantly alleviates the mapping challenge in face restoration.
- We introduce ART-FR, a promising face restoration method that employs a masked auto-regressive model, complemented by an innovative architecture designed to integrate LQ condition during the generative procedure. To the best of our knowledge, this is the first work to leverage Auto-Regressive Transformer for image restoration task.
- We assess the performance of ART-FR through comprehensive experiments, demonstrating its efficacy with both superior quantitative metrics and qualitative evaluations.

## 2 RELATED WORK

### 2.1 BLIND FACE RESTORATION.

Blind face restoration, a specialized subfield of image restoration, has garnered significant attention in recent years. Given the inherently ill-posed nature, the restoration process typically necessitates auxiliary priors to regularize the solution space.

**Previous works.** Traditional approaches primarily relied on two types of face-specific priors: geometric priors and reference priors. Methods based on geometric priors, including facial landmarks (Chen et al., 2018), face parsing maps (Chen et al., 2021), and 3D shapes (Hu et al., 2020), struggle to accurately estimate geometric information from degraded LQ images. Reference-based approaches (Dogan et al., 2019; Li et al., 2020) use images of the same identity, which could be difficult to obtain in practice. Leveraging pre-trained generative models, such as StyleGAN (Karras et al., 2019) and DDPM (Ho et al., 2020), the generative priors can restore rich facial details and deliver plausible performance. GAN inversion (Gu et al., 2020; Menon et al., 2020) aim to identify the most compatible latent code in GAN's latent space based on the LQ input. GPEN (Yang et al., 2021) and GFPGAN (Wang et al., 2021) incorporate additional structural information from LQ images. Wang et al. (2023b) and Lin et al. (2023) employ diffusion models as enhancement modules to restore high-quality facial details from coarse restoration outputs.

**Codebook priors.** Codebook-based methods utilize a VQGAN to learn a codebook that captures rich texture information for face restoration. VQFR (Gu et al., 2022) uses a parallel decoder to directly integrate LQ information into the decoding process, while Codeformer (Zhou et al., 2022) introduces an additional prediction network to match LQ images with corresponding codes from the codebook. However, these methods still have limitations as they overlook the accurate modeling of the mapping from LQ features to the discrete representations of the HQ images, leading to unrealistic facial details. In contrast, our approach focuses on mitigating this particular challenge.

### 2.2 AUTO-REGRESSIVE TRANSFORMER

**Image Tokenizer and Auto-Regression.** Image tokenizer functions analogous to WordPiece algorithms in language models, effectively encoding 2D images into 1D token sequences. VQVAE (Van Den Oord et al., 2017) employs an encoder-decoder framework that quantizes patch-level features to the nearest entry in a learned codebook through a self-reconstruction technique. VQGAN (Esser et al., 2021) advances VQVAE by integrating adversarial and perceptual losses, resulting in more realistic image reconstructions. ViT-VQGAN (Yu et al., 2021) further improves upon this by replacing the CNN-based encoder-decoder with a Vision Transformer, achieving superior reconstruction quality.

As a downstream task in image quantization, auto-regressive models are tasked with modeling distribution of the discrete token sequences. VQVAE employs PixelCNN (Van Den Oord et al., 2016) to autoregressively capture the dependencies between image tokens. iGPT (Chen et al., 2020) pioneers the application of the powerful Transformer architecture for image generation. More recently, RQVAE (Lee et al., 2022) and VAR (Tian et al., 2024) introduce novel autoregressive frameworks that diverge from the standard raster-scan approach to a coarse-to-fine generation strategy.

**Masked Auto-Regressive models.** Masked Auto-Regressive models originate from BERT (Devlin, 2018) in Masked language modeling task. MaskGIT (Chang et al., 2022) introduces the Masked prediction framework to image generation for the first time. MagViT (Yu et al., 2023a) and its successor, MagViT-2 (Yu et al., 2023b), extend this approach to video generation. MUSE (Chang et al., 2023) scales up MaskGIT's architecture to 3 billion parameters, achieving remarkable text-to-image synthesis.

# 3 METHOD

The core idea of this work is to reformulate the image restoration problem as a conditional generation problem in discrete latent space with lower complexity. The inherent characteristics of image discretization allow us to leverage the powerful auto-regressive generative model to bridge the gap between LQ images and HQ latent representations. An overview of ART-FR is illustrated in Fig. 2.

The training of ART-FR can be divided into two distinct stages. In stage I, we pre-train a VQGAN to obtain discrete representations of the training data, as well as a high-quality facial texture bank (Sec. 3.1). For the stage II, we develop a Masked Visual Token Transformer that conditioned on LQ features to model the distribution of the discrete token sequences corresponding to the training data (Sec. 3.2). The details of the sampling procedure during inference is illustrated in Sec. 3.3.

## 3.1 TRAINING STAGE I: DISCRETE REPRESENTATION LEARNING

In stage I, HQ image pixels $\boldsymbol{I}_h \in \mathbb{R}^{H \times W \times 3}$ are tokenized into a discrete representation $\boldsymbol{q} \in \mathbb{Q}^{h \times w}$ for subsequent auto-regressive modeling of the latent space distribution. Here, $H/h$ (or $W/w$) represents the downsampling ratio of the image tokenizer. During the parallel procedure, a comprehensive and context-rich codebook prior is learned, effectively encoding high-quality facial details.

**Image Tokenizer Architecture.** We utilize an encoder-quantizer-decoder architecture that mirrors VQGAN. For an HQ input $\boldsymbol{I}_h$, the encoder $E_h$ projects $\boldsymbol{I}_h$ to its feature map $\boldsymbol{f}_h \in \mathbb{R}^{h \times w \times d}$, where $d$ denotes the dimension of feature vectors. The quantizer encompasses a codebook $\boldsymbol{Z} \in \mathbb{R}^{K \times C}$, with $K$ and $C$ representing the size and dimension of codebook respectively. The quantization process maps each feature vector $f_h^{(i,j)}$ to the code $q^{(i,j)}$ corresponding to the closest element $z^{(i,j)}$ within the codebook. From this process, we can obtain a discrete code sequence $\boldsymbol{q}$ corresponding to $\boldsymbol{I}_h$,

$$\boldsymbol{f}_h = E_h(\boldsymbol{I}_h); \quad q^{(i,j)} = \arg\min_k ||f_h^{(i,j)} - z_k||_2, \ z_k \in \boldsymbol{Z}. \tag{1}$$

During the decoding phase, the code indices $\boldsymbol{q}$ is reassigned back to the feature vector $\hat{\boldsymbol{f}}_h \in \mathbb{R}^{h \times w \times d}$, after which the decoder, $D_h$, reconstructs the image pixels from these feature vectors.

$$\hat{f}_h^{(i,j)} = z_{q^{(i,j)}}; \quad \boldsymbol{I}_h^{rec} = D_h(\hat{\boldsymbol{f}}_h). \tag{2}$$

**Training Losses.** Since the quantization is non-differentiable, a straight-through gradient estimator, $\hat{\boldsymbol{f}} = sg[\hat{\boldsymbol{f}} - \boldsymbol{f}] + \boldsymbol{f}$, is used to propagate the gradient from the decoder to the encoder, where $sg[\cdot]$ is a stop-gradient operation. Specifically, for codebook learning

$$\mathcal{L}_{VQ} = ||sg[\boldsymbol{f}_h] - \hat{\boldsymbol{f}}||_2^2 + \beta||\boldsymbol{f}_h - sg[\hat{\boldsymbol{f}}]||_2^2, \tag{3}$$

where the second term pushes the feature vectors closer to the codebook entries. Here, $\beta = 0.25$ serves as a balancing factor that modulates the relative update rates of the encoder and the codebook. For reconstruction learning,

$$\mathcal{L}_{rec} = \mathcal{L}_2(\boldsymbol{I}_h, \boldsymbol{I}_h^{rec}) + \mathcal{L}_P(\boldsymbol{I}_h, \boldsymbol{I}_h^{rec}) + \lambda_G \mathcal{L}_G(\boldsymbol{I}_h, \boldsymbol{I}_h^{rec}), \tag{4}$$

where $\mathcal{L}_2(\cdot)$ is a fidelity loss in pixel level, $\mathcal{L}_P(\cdot)$ is a perceptual loss from LPIPS, $\mathcal{L}_G(\cdot)$ is an adversarial loss, and $\lambda_G = 0.8$ in our setting.

## 3.2 TRAINING STAGE II: CONDITIONAL MASKED VISUAL TOKEN MODELING

Leveraging the discrete representations (HQ tokens) derived from Stage I, we reformulate face restoration as a conditional generation task within discrete latent space. we propose to learn a

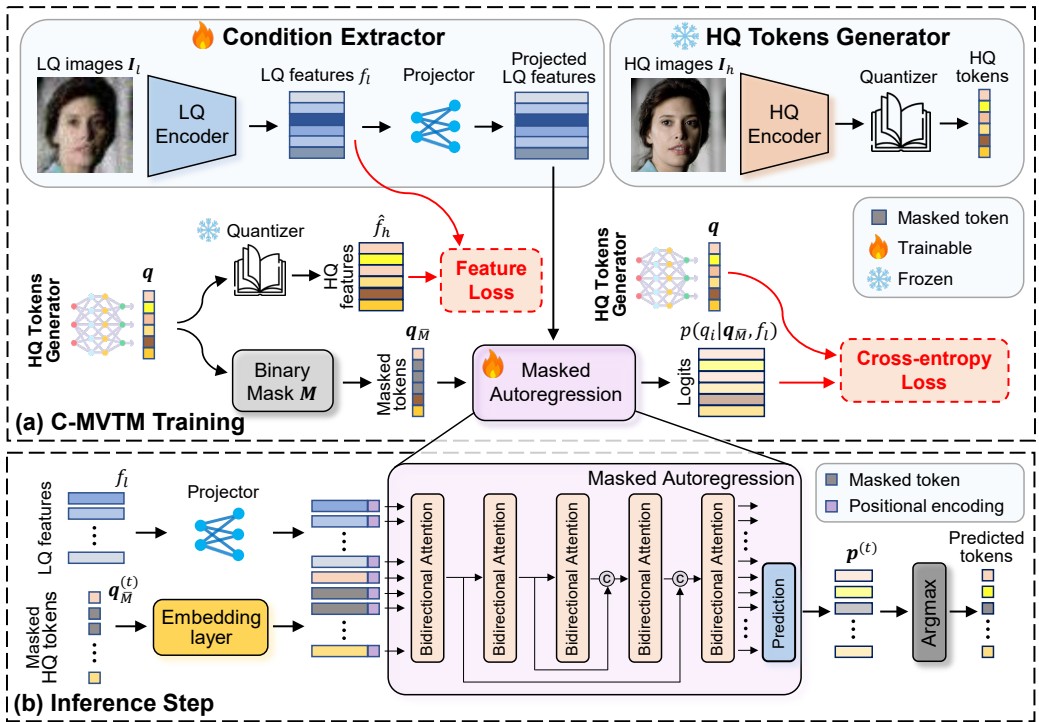

Figure 3: **Training framework and inference step of ART-FR. (a)** ART-FR are trained under two levels. At the feature level, ART-FR trains an LQ encoder to minimize the discrepancy between LQ and HQ features. At the code level, an additional masked auto-regressive model is trained to predict the ground truth tokens from HQ images. The codebook and HQ decoder are fixed during training phase. **(b)** In each inference step, LQ features and HQ tokens are jointly input into the auto-regressive model. New tokens are generated based on the output logits, with sampling performed at the position of highest confidence. The network architecture incorporates a bidirectional attention mechanism and utilizes long skip connections to enhance information transfer. LQ features are used as prefix tokens to guide the auto-regressive generation process.

**Condition Extractor**, including an LQ encoder and a linear projector, along with a **Masked Auto-Regressive Model** for Conditional Masked Visual Token Modeling (C-MVTM), as illustrated in Fig. 3 (a). During Stage II training, the HQ encoder and quantizer jointly function as the **HQ Tokens Generator**, while the quantizer and decoder are fixed to maintain the high-quality image context.

**C-MVTM Training.** Before auto-regressive modeling, we employ the HQ Token Generator to produce the modeling target $q \in \mathbb{R}^{h \times w}$, which is subsequently flattened into a latent tokens sequence, denoted as $q = [q_i]_{i=1}^{h \cdot w}$. Additionally, we define $M = [m_i]_{i=1}^{h \cdot w}$ as a binary mask. A subset of tokens is selected to be masked and replaced with the marker [MASK]. Specifically, a token $q_i$ is substitute with [MASK] if $m_i = 1$. Conversely, when $m_i = 0$, the token remains unaltered.

Given that the quality of generation in C-MVTM is significantly influenced by the mask ratio during training, we employ a concave mask scheduling function $\gamma(r) = cos(\pi/2 * r) \in (0, 1]$ consistent with the principles of MaskGIT (Chang et al., 2022). The function $\gamma(\cdot)$ follows a less-to-more process. Initially, the majority of tokens are masked, allowing the model to concentrate on making a limited number of accurate predictions with high confidence. As training progresses, the mask ratio decreases substantially, compelling the model to produce a significantly larger number of correct predictions. During the mask operation, we first sample a uniform distribution r ranging from 0 to 1, and then select $(\gamma(r) \cdot h \cdot w)$ tokens for replacement with the [MASK].

We denote $q_{\bar{M}}$ as the remaining tokens after applying $M$ to $q$. During C-MVTM procedure (Fig. 3 (a)), we condition on $f_l$ and input $q_{\bar{M}}$ into the bidirectional transformer to predict the distribution of masked tokens $p(q_i|q_{\bar{M}}, f_l)$, $m_i = 1$, and then calculate the cross-entropy between the ground-truth one-hot token and the predicted tokens.

**Bidirectional Transformer Architecture.** Follow Bao et al. (2023), we employ a ViT-based architecture for token prediction. In practice, we treat all inputs, including LQ features, HQ tokens,

and [MASK] tokens, as tokens of equal status, and apply long skip connections between the shallow and deep layers, as shown in Fig. 3 (b). Our architecture is characterized by the following aspects: (1) For the combination of the long skip branch, letting $\boldsymbol{h}_m, \boldsymbol{h}_s$ be the feature maps from main branch and the skip branch, we concatenate them along channel dimension and perform a linear projection, i.e., $Linear(Concat(\boldsymbol{h}_m, \boldsymbol{h}_s))$. (2) To incorporate the LQ condition, we use a linear projector to ensure it matches the dimension of the token embedding, and then prepend it as prefix tokens before the [MASK] tokens. (3) For position embedding, we utilize a 1-D learnable position embedding, consistent with the original ViT architecture.

**Training Losses.** To accurately model the mapping from LQ features to HQ tokens, we incorporate losses at both global and local levels. For the global content level, we employ an L2 loss (Feature Loss in Fig. 3 (a)) to reduce the domain discrepancy between HQ and LQ images within the feature maps, thereby ensuring that they exhibit similar content and semantic structures,

$$\boldsymbol{f}_l = E_l(\boldsymbol{I}_l); \quad \mathcal{L}_{feat} = ||\boldsymbol{f}_l - sg(\hat{\boldsymbol{f}}_h)||_2^2. \tag{5}$$

For local texture refinement, we minimize the negative log-likelihood (Cross-entropy Loss in Fig. 3 (a)) of masked tokens to achieve precise correction of facial details,

$$\mathcal{L}_{mask} = -\mathbb{E}_{\boldsymbol{q} \in \mathcal{D}}\Big[\sum_{\forall i \in [1, h \cdot w], m_i = 1} log\, p(q_i|\boldsymbol{q}_{\bar{\boldsymbol{M}}}, \boldsymbol{f}_l)\Big]. \tag{6}$$

### 3.3 Inferencing: Iterative Decoding Refinement

The overall inference process is depicted in Fig. 2. The prediction starts with a sequence where every position is initialized as [MASK], denoted as $\boldsymbol{q}^{(0)}$. The masked tokens are progressively refined into a HQ token sequence through an iterative prediction process. Additionally, multiple tokens can be generated in a single inference step, thereby reducing the number of generation steps. In our implementation, the decoding algorithm constructs an image over $T = 8$ steps.

At single inference step $t$ (Fig. 3 (b)), the predicted tokens and LQ features are first mapped to the same embedding space, and then fed into the Masked Auto-Regressive model. The model first predicts the distribution $\boldsymbol{p}^{(t)} \in \mathbb{R}^{N \times K}$ of all tokens and refer the prediction score as the confidence measure of each token. In masked-token prediction, we allow the tokens to be predicted based on the information from both sides of the token sequence using a bidirectional attention mechanism. In the second step, we sample $n = \gamma(t/T) \cdot N$ tokens from the masked positions with most confident predictions, where $N$ is the total sequence length and $T$ is the number of iterations. Finally, we update the mask $\boldsymbol{M}^{(t+1)}$ from $\boldsymbol{q}^{(t+1)}$ for subsequent iterations.

Conceptually, this Masked-token Prediction process can be written as the modeling of conditional distribution $p(q_{i+1}^s, q_{i+2}^s, ..., q_j^s | q_1^s, q_2^s, ..., q_i^s, \boldsymbol{f}_l)$, where $(q_1^s, q_2^s, ..., q_N^s)$ is the shuffled version of $(q_1, q_2, ..., q_N)$. And the generation process can be written as:

$$p_\theta(q_1, q_2, ..., q_N | f_l) = p_\theta(\boldsymbol{Q}_1, \boldsymbol{Q}_2, ..., \boldsymbol{Q}_T | \boldsymbol{f}_l) = \prod_{k=1}^{T} p_\theta(\boldsymbol{Q}_k | \boldsymbol{Q}_1, \boldsymbol{Q}_2, ..., \boldsymbol{Q}_{k-1} | \boldsymbol{f}_l), \tag{7}$$

where $\boldsymbol{Q_k} = \{q_{i+1}^s, q_{i+2}^s, ..., q_j^s\}$ is the predicted tokens in k-th step.

## 4 Experiments, Results and Discussions

### 4.1 Experimental setting

**Implementation.** During the VQGAN training phase, a downsampling rate of 32 was applied, meaning that an image of size $512 \times 512 \times 3$ is converted into a $16 \times 16$ (256) token sequence. The size of the codebook was set to 1024. In the auto-regressive modeling phase, the total number of sampling steps was set to 8. The Adam (Kingma, 2014) optimizer was used throughout the training process with a batch size of 24. The learning rate was set to $8 \times 10^{-5}$ for Stage I and was decayed from $2 \times 10^{-4}$ to $2 \times 10^{-5}$ using a cosine annealing schedule for Stage II. The two stages were trained with 500K and 150K iterations, respectively. Our method was performed on the PyTorch with 4 NVIDIA GeForce RTX 4090 GPUs.

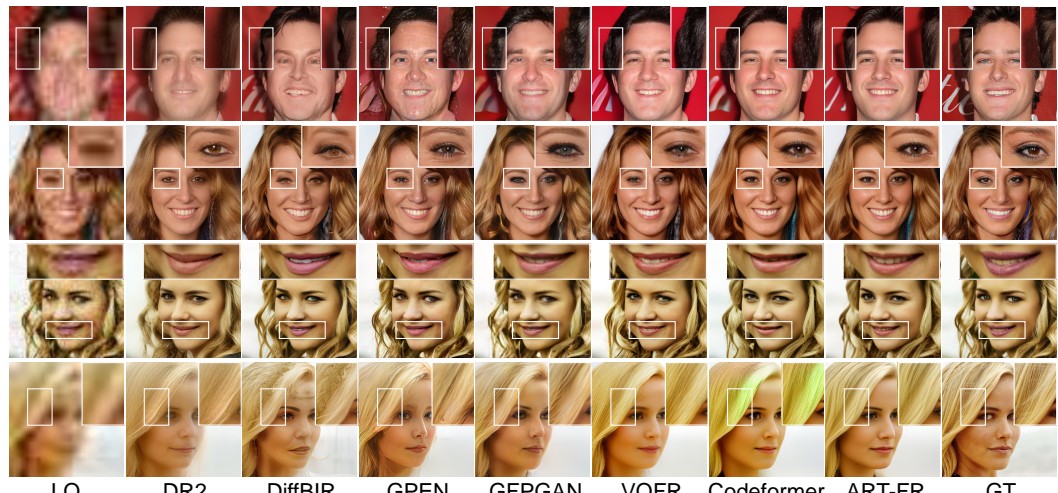

LQ      DR2      DiffBIR      GPEN      GFPGAN      VQFR      Codeformer      ART-FR      GT

Figure 4: **Qualitative comparison on the CelebA-Test.** ART-FR is capable of robustly recovering high-quality facial details and components, such as hair and mouth, even under severe degradation.

**Training Datasets.** The ART-FR model was trained on the FFHQ dataset, which contains 70,000 high-quality facial images. All images were resized to $512 \times 512$ for training. Following previous work(Yang et al., 2021; Zhou et al., 2022; Wang et al., 2021), we generated degraded facial images using the degradation model:

$$I_l = \{[(I_h \otimes k_\sigma) \downarrow_r + n_\delta]_{\mathrm{JPEG}_q}\} \uparrow_r, \tag{8}$$

where $\sigma, r, \delta$ and $q$ are randomly sampled from $[1, 15], [1, 30], [0, 20]$ and $[40, 100]$, respectively.

**Testing Datasets.** We evaluate ART-FR using one synthetic dataset and three real-world datasets. The synthetic dataset, named CelebA-Test, comprises 3,000 high-quality images sourced from CelebA-HQ (Karras, 2017) , with the corresponding low-quality images generated according to Eq. (8). For the real-world datasets, we use three representative ones, LFW-Test (Huang et al., 2008), WIDER-Test (Yang et al., 2016), and CelebChild-Test (Wang et al., 2021), to test the generalization ability. LFW-Test contains 1,711 mildly degraded face images in the wild, with one image per individual from the LFW dataset. WIDER-Test includes 970 heavily degraded face images selected from the WIDER Face dataset. CelebChild-Test comprises 180 child faces of celebrities collected from the Internet.

**Evaluation Metrics.** To evaluate ART-FR's performance on the CelebA-Test dataset with ground truth, we employ PSNR, and LPIPS (Zhang et al., 2018) as evaluation metrics. For real-world datasets without ground truth, we use the commonly applied non-reference perceptual metrics, FID (Heusel et al., 2017) and NIQE (Mittal et al., 2012). Additionally, embedding angle of ArcFace (Deg) and landmark distance (LMD) are used to assess the retention of original facial features (Deng et al., 2019).

**Extensional applications.** In this work, we further extended our model from the basic blind image restoration task to image colorization, inpainting, old photo restoration, and super-resolution with high downsampling rates, as illustrated in Figure 1. For each task, we fine-tuned or retrained our model. The results demonstrate that ART-FR is applicable to a wide range of tasks and exhibits strong robustness under severe degradations, such as large-area occlusions and high downsampling rates. More results of the extensions are presented in the Appendix C.1.

### 4.2 COMPARISONS WITH STATE-OF-THE-ART METHODS

We compare ART-FR with several state-of-the-art methods, encompassing two diffusion-based methods, DR2 (Wang et al., 2023b) and DiffBIR (Lin et al., 2023), two GAN-based methods, GFPGAN (Wang et al., 2021) and GPEN (Yang et al., 2021), as well as two VQ-based techniques, Codeformer (Zhou et al., 2022) and VQFR (Gu et al., 2022).

Figure 5: **Qualitative comparisons on three real-world datasets.** ART-FR demonstrates realistic reconstruction effects, *e.g.*, skin, while maintaining high fidelity, *e.g.*, the preservation of the glasses in the second row.

Table 1: **Quantitative comparisons.** Red and blue indicate the best and second-best, respectively.

(a) The *synthetic* CelebA-Test dataset.

| Metrics Methods | Realness metrics | | | Fidelity metrics | | |
|---|---|---|---|---|---|---|
| | FID↓ | NIQE↓ | LPIPS↓ | Deg↓ | LMD↓ | PSNR↑ |
| Input | 255.57 | 17.881 | 0.593 | 76.21 | 13.21 | 21.82 |
| DR2 | 48.80 | 5.417 | 0.460 | 66.72 | 6.54 | 21.37 |
| DiffBIR | 40.59 | 5.371 | 0.506 | 62.16 | 9.91 | 21.06 |
| GPEN | 33.24 | 5.580 | 0.436 | 59.88 | 5.99 | 21.40 |
| GFPGAN | 21.74 | 4.348 | 0.432 | 58.07 | 5.14 | 21.68 |
| CodeFormer | 27.47 | 4.913 | 0.401 | 57.24 | 4.80 | 21.25 |
| VQFR | 20.55 | 4.645 | 0.410 | 58.38 | 4.98 | 21.45 |
| **ART-FR (Ours)** | 25.35 | 4.769 | 0.393 | 57.54 | 4.68 | 21.75 |

(b) *Real-world* datasets.

| Datasets Methods | LFW-Test | | WIDER-Test | | CelebChild | |
|---|---|---|---|---|---|---|
| | FID↓ | NIQE↓ | FID↓ | NIQE↓ | FID↓ | NIQE↓ |
| Input | 137.56 | 11.214 | 202.06 | 13.498 | 144.42 | 9.17 |
| DR2 | 53.55 | 4.734 | 54.82 | 5.318 | 137.03 | 4.930 |
| DiffBIR | 61.71 | 5.836 | 55.40 | 5.574 | 125.39 | 5.483 |
| GPEN | 55.09 | 4.454 | 59.48 | 5.811 | 109.13 | 4.711 |
| GFPGAN | 48.66 | 4.478 | 42.98 | 4.380 | 120.89 | 4.757 |
| CodeFormer | 52.40 | 4.446 | 37.66 | 4.466 | 124.34 | 4.954 |
| VQFR | 50.94 | 3.821 | 38.89 | 4.036 | 117.92 | 4.411 |
| **ART-FR (Ours)** | 46.75 | 4.302 | 45.47 | 3.888 | 110.36 | 4.087 |

**Evaluation on Synthetic Datasets.** The quantitative results are presented in Table 1 (a). For metrics with explicit reference images, including LPIPS, Deg, LMD, and PSNR, ART-FR achieved state-of-the-art performance compared to previous methods. ART-FR exhibits the lowest LPIPS score, indicating that our method achieves the highest level of perceptual quality. Regarding the FID and NIQE metrics, although ART-FR only ranks third, it exhibits enhanced generalization to real datasets. This suggests that our approach is capable of generating facial images with realistic details that closely resemble real human faces. For fidelity, our method outperformed previous models comprehensively. ART-FR achieves the highest PSNR and LMD scores, highlighting its ability to preserve identity and accurately recover facial expressions and fine details.

Qualitative results are presented in Fig. 4, 11. ART-FR is capable of producing plausible results with both realness and fidelity even under severe degradation, as evidenced by the hair in the first row. In contrast, diffusion-based methods often yield over-smoothed results and introduce spurious details when dealing with severe degradation; GAN-based approaches frequently produce noticeable artifacts; while other VQ-based methods struggle to accurately integrate degraded information and predict authentic HQ tokens, as evidenced by the erroneous hair color prediction in the fourth row. To further substantiate the robustness of the ART-FR model when confronted with complex degradation issues, we conducted additional experiments, the results of which are detailed in Appendix B.

**Evaluation on Real-world Datasets.** ART-FR was evaluated across three real-world test datasets, with the quantitative outcomes detailed in Table 1 (b). For perceptual quality, measured by the NIQE metric, our method achieved the highest scores on both the Wider-Test and CelebAChild-Test datasets, surpassing the second place by a large margin. Additionally, ART-FR secured the second position on the LFW-Test dataset. In terms of the FID metric, ART-FR delivered the best performance on the LFW-Test dataset and demonstrated competitive result on the CelebChild dataset. As for the LFW-Test, which has the most severe degradation levels, the two best-performing methods, Codeformer and VQFR, both used skip connections between the encoder and decoder. This implies that multi-scale LQ information is beneficial for face restoration under severe degradation.

Visual comparisons presented in Fig. 5 illustrate that ART-FR can generate highly detailed facial features, outperforming previous methods. In terms of fidelity, ART-FR effectively preserves key facial characteristics from degraded images, such as the glasses in the second row. This success is due to the incorporation of degradation conditions in our approach, which effectively guides the token generation throughout the restoration process. We provide more visual comparisons in Fig. 12.

Table 2: **Ablation studies of ART-FR on the CelebA-Test.** We conducted ablation studies in two key aspects: network architecture and loss function. The terms "Skip connection" and "AR module" refer to the long skip connections between bidirectional transformers and our auto-regressive token prediction module, respectively.

| Exp. | Skip connection | AR module | Feature loss | LPIPS↓ | LMD↓ | Deg↓ |
|------|-----------------|-----------|--------------|--------|------|------|
| (a) | | | ✓ | 0.4185 | 5.1325 | 59.9775 |
| (b) | | ✓ | ✓ | 0.3965 | 4.9041 | 58.1013 |
| (c) | ✓ | ✓ | | 0.4050 | 5.2700 | 59.7662 |
| (d) | ✓ | ✓ | ✓ | **0.3943** | **4.7256** | **57.7465** |

Table 3: **The flexible introduction of LQ condition.** We conducted three methods for integrating LQ conditions: adding a Cross-attention module to the auto-regressive model, concatenating mask embeddings with LQ features, and a prefix token strategy.

| Methods | LPIPS↓ | LMD↓ | Deg↓ | PSNR↑ |
|---------|--------|------|------|-------|
| Cross-attention | 0.3949 | 4.7667 | 57.9144 | 21.7019 |
| Concatenation | 0.3974 | 4.8648 | 58.7128 | 21.6129 |
| Prefix token | 0.3943 | 4.7256 | 57.7465 | 21.7309 |

## 4.3 ABLATION STUDY

**Architecture.** In this study, we first examined the efficacy of auto-regressive token prediction on image restoration task. We conducted a comparative analysis between ART-FR and two other token prediction based methods, *i.e.*, Codeformer and VQGAN. As illustrated in Fig. 6, ART-FR demonstrated a significantly higher accuracy rate in token prediction across various degradation levels compared to the other two methods. Additional comparative analyses were performed in Table 2 (Exp. (a) and (d)). The results demonstrate that auto-regressive token prediction is highly beneficial in enhancing the perceptual quality and fidelity of face restoration. We have incorporated long skip connections in the architecture of ART-FR to alleviate the difficulty of information transmission.

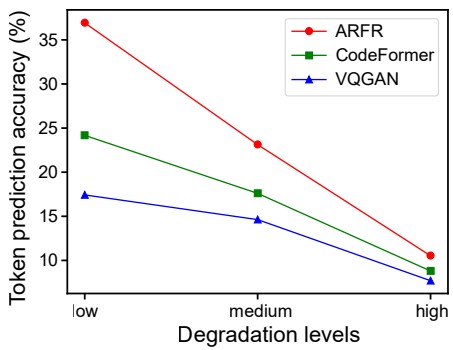

Figure 6: **Token prediction accuracy.** Accuracy comparison on various levels of degradation.

To substantiate the efficacy of this design, comparative experiments were conducted in Exp. (b) and (d) as depicted in Table 2. The superior performance further substantiates the efficacy of this architectural improvement.

**Feature Loss.** We introduced a feature loss function to minimize the distance between LQ and HQ feature maps, with the aim of preliminarily removing the degradation from the LQ images and thereby reducing the complexity of subsequent restoration. This analysis is presented in Exp. (c) and (d) as detailed in Table 2. The improved performance on LPIPS, LMD, and Deg demonstrates that the incorporation of feature loss endows the LQ encoder with a certain degree of degradation removal capability, thereby facilitating the process of face restoration. We provide an extended visual comparison of the aforementioned methods in Appendix Fig. 10.

**LQ condition.** To demonstrate the versatility of the ART-FR framework. We implement three distinct methods for incorporating LQ conditions: The first approach integrates a cross-attention module to incorporate conditional information, a strategy extensively applied in conditional generative models (Rombach et al., 2022). The second method concatenates LQ features with masked token embeddings along the channel dimension at the input of the auto-regressive model. The final approach introduces LQ features as prefix tokens. For specific implementation, refer to Appendix Fig. 14. The result are presented in Table 3. ART-FR demonstrates strong performance across all three conditions, achieving favorable results in both perceptual and fidelity metrics.

## 5 CONCLUSION

In this paper, we propose ART-FR to address the problem of face restoration by reformulating it as a conditional generative task within a discrete latent space. Drawing inspiration from language models and auto-regressive image generation, we employed the discrete distribution modeling capabilities of the Auto-regressive Transformer to map LQ images to HQ tokens, which are then translated back into the image domain. Our method, encompassing Masked Visual Token Modeling, a tailored auto-regressive architecture, and the strategic incorporation of LQ conditions, has synergistically enhanced the efficiency and robustness of face restoration.

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

## A    MORE IMPLEMENTATION DETAILS

### A.1    NETWORK STRUCTURE

**Visual Tokenizer Structures.** We adopt the same CNN basic structure as VQGAN (Esser et al., 2021). The encoder of ART-FR utilizes six residual blocks with channel dimensions of [64, 128, 128, 256, 256, 512]. The first five residual blocks are followed by a downsample module to reduce the image size from $512 \times 512$ to $16 \times 16$. When the image is downsampled to a resolution of 16, an attention module is introduced to fully integrate global features. Similarly, the decoder of ART-FR employs six residual blocks with channel dimensions of [512, 256, 256, 128, 128, 64] to upsample the image back to its original resolution. The quantizer trains a codebook $\boldsymbol{Z} \in \mathbb{R}^{1024 \times 256}$ to restore facial texture information. A linear projector is added between the quantizer and the encoder (or decoder) to align the dimensions.

**Masked Auto-regression Structures.** Our masked auto-regressive model (training stage I) utilizes a BERT-like architecture, which employs multiple layers of bidirectional Transformers to encode the relationships between tokens in both directions. In our setting, 512 tokens are fed into the bidirectional Transformers. The first 256 tokens, generated by the LQ encoder, serve as the LQ condition to guide the generation process, while the remaining 256 tokens are masked HQ tokens, with the masked parts replaced by a special token [MASK]. The model learns both a set of word embeddings and position embeddings with dimension 768 to encode the token indices and their positions, enabling the model to capture the distribution of the token sequence. The input to the Transformers can be represented by

$$\textbf{Input} = \textbf{Embed}_{words}(token\_ids) + \textbf{Embed}_{position}(position\_ids). \tag{9}$$

Between the Transformer layers, shallow features $\boldsymbol{h}_s$ and deep features $\boldsymbol{h}_m$ are fused by $Linear(Concat(\boldsymbol{h}_m, \boldsymbol{h}_s))$. At the model's output, a prediction head is applied to estimate the distribution of the positions where [MASK] located.

### A.2    EXTENTION EXPERIMENTS SETTING

To achieve face inpainting and colorization, we fine-tune our pre-trained model for 20,000 iterations using the Adam optimizer with a batch size of 24. In the construction of the inpainting dataset, we generate LQ data by drawing random irregular polyline masks, following the methodology of GPEN (Yang et al., 2021). For the colorization dataset, we employ random color jittering and grayscale conversion as GFPGAN(Wang et al., 2021). We developed a super-resolution model with 200K iterations from scratch. As for old photo restoration, we utilized a pre-trained version of ART-FR without the need for additional training.

## B    ROBUSTNESS OF ART-FR TO DEGRADATIONS

In this section, we present an analysis demonstrating the robustness of the ART-FR model across a wide spectrum of image degradations. In this study, we validate the performance of our model on the task of denoising. For the generation of LQ images, during each iteration, we randomly select a noise level from the set 25, 50, 75, 100 to introduce into the HQ images. To test the limits of the robustness of the ART-FR model, we further escalate the noise levels to 125, 150, 175, 200. We denote the models trained under these two settings as Model A and Model B, respectively.

To better evaluate the quality of image restoration, we include additional no-reference image quality assessment metrics: MUSIQ (Ke et al., 2021), MANIQA (Yang et al., 2022), and CLIPIQA (Wang et al., 2023a). To verify the robustness of ART-FR, which leverages an auto-regressive model for

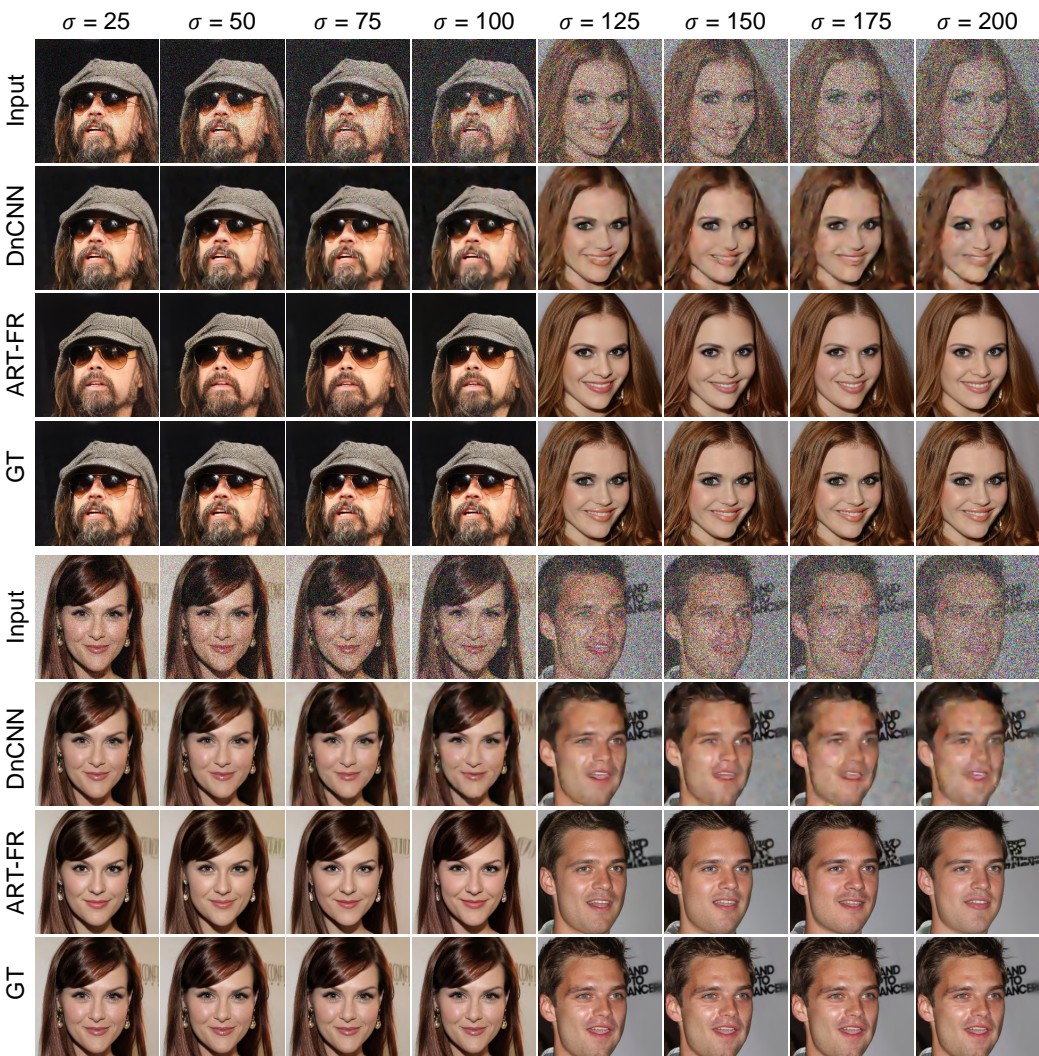

$\sigma = 25$  $\sigma = 50$  $\sigma = 75$  $\sigma = 100$  $\sigma = 125$  $\sigma = 150$  $\sigma = 175$  $\sigma = 200$

Figure 7: **Qualitative comparison between ART-FR and DnCNN.** ART-FR is capable of producing high-quality restoration results under various severe degradation conditions.

image restoration, we compare it with a traditional method, DnCNN (Zhang et al., 2017), which is trained in an end-to-end manner. For experimental validation, we selected the first 100 images from the CelebA-Test dataset as the testset. The experimental results are depicted in Fig. 7 and Table 4.

As shown in Table 4, ART-FR significantly outperforms DnCNN in terms of image quality (both on realness and aesthetics) across all levels of degradation, from mild to extreme. On reference-based metrics, ART-FR gradually surpasses DnCNN as the degradation level increases. In the longitudinal comparison of a single model, ART-FR exhibits stable performance across all metrics as degradation intensifies, whereas DnCNN shows a sharp decline in performance. In terms of the LPIPS, as the noise level increases from 25 to 100, ART-FR's LPIPS score rises from 0.320 to 0.338 (+0.018), while DnCNN's score jumps from 0.247 to 0.420 (+0.173), representing a nearly 10× larger increase compared to ART-FR. From the qualitative results in Fig. 7, even under extreme degradation, ART-FR still produces plausible restoration results. In contrast, DnCNN generates smoother outputs, which are less favorable. ART-FR's stable performance on no-reference metrics (Table 4) further supports this observation.

Furthermore, we investigated the performance of ART-FR when handling degradations that deviate from the training degradation domain. Specifically, we evaluated Model A on the noise settings of Model B. The results are presented in Fig. 8 and Table 5. From the qualitative analysis, ART-FR still produces high-quality restoration results on degraded faces outside the domain of the training degradation settings. As shown in Table 5, our model outperforms DnCNN across all metrics (except

Table 4: **Quantitative comparison between ART-FR and DnCNN.** The models were trained across a wide range of degradation levels. The results indicate that the auto-regressive based face restoration method demonstrates greater robustness when handling extensive degradations.

| Model | Noise level | Metrics Methods | Reference Deg↓ | LMD↓ | PSNR↑ | FID↓ | LPIPS↓ | w/o Reference NIQE↓ | MUSIQ↑ | CLIPIQA↑ | MANIQA↑ |
|---|---|---|---|---|---|---|---|---|---|---|---|
| A | 25 | DnCNN | **14.32** | **0.60** | **34.08** | **43.38** | **0.247** | 5.48 | 70.46 | 0.558 | 0.438 |
|   |   | ART-FR | 39.33 | 2.46 | 24.07 | 56.71 | 0.320 | **4.68** | **74.27** | **0.660** | **0.534** |
|   | 50 | DnCNN | **22.45** | **1.08** | **31.28** | 71.62 | 0.331 | 6.54 | 65.98 | 0.469 | 0.385 |
|   |   | ART-FR | 40.32 | 2.45 | 24.08 | **59.45** | **0.327** | **4.67** | **74.35** | **0.661** | **0.535** |
|   | 75 | DnCNN | **28.74** | **1.59** | **29.57** | 94.31 | 0.383 | 7.36 | 59.53 | 0.384 | 0.328 |
|   |   | ART-FR | 41.33 | 2.51 | 24.02 | **60.37** | **0.332** | **4.69** | **74.52** | **0.662** | **0.538** |
|   | 100 | DnCNN | **35.51** | **2.11** | **28.30** | 110.62 | 0.420 | 7.778 | 53.54 | 0.334 | 0.279 |
|   |   | ART-FR | 42.41 | 2.64 | 23.95 | **63.31** | **0.338** | **4.69** | **74.59** | **0.658** | **0.535** |
| B | 125 | DnCNN | **40.43** | **2.46** | **27.53** | 114.76 | 0.435 | 8.04 | 47.64 | 0.309 | 0.249 |
|   |   | ART-FR | 44.22 | 2.75 | 23.85 | **64.91** | **0.345** | **4.75** | **74.58** | **0.660** | **0.536** |
|   | 150 | DnCNN | **44.67** | 2.86 | **26.79** | 125.90 | 0.454 | 8.19 | 44.52 | 0.284 | 0.220 |
|   |   | ART-FR | 45.52 | **2.84** | 23.71 | **65.54** | **0.350** | **4.80** | **74.79** | **0.660** | **0.537** |
|   | 175 | DnCNN | 49.35 | 3.48 | **26.14** | 142.70 | 0.474 | 8.36 | 40.13 | 0.260 | 0.190 |
|   |   | ART-FR | **46.07** | **2.96** | 23.58 | **68.37** | **0.354** | **4.77** | **74.58** | **0.661** | **0.539** |
|   | 200 | DnCNN | 53.20 | 3.79 | **25.55** | 154.38 | 0.490 | 8.45 | 36.33 | 0.245 | 0.163 |
|   |   | ART-FR | **47.28** | **3.11** | 23.46 | **68.26** | **0.358** | **4.73** | **74.71** | **0.663** | **0.538** |

PSNR) while maintaining highly stable performance. Notably, the FID score remains unaffected by varying levels of degradation. This is because our auto-regressive based approach learns the prior distribution of the data during training.

Table 5: A **quantitative** comparison between ART-FR and DnCNN was conducted under severe degradation conditions that deviate from the training domain.

| Model | Noise level | Metrics Methods | Reference Deg↓ | LMD↓ | PSNR↑ | FID↓ | LPIPS↓ | w/o Reference NIQE↓ | MUSIQ↑ | CLIPIQA↑ | MANIQA↑ |
|---|---|---|---|---|---|---|---|---|---|---|---|
| A | 125 | DnCNN | **40.88** | **2.65** | **27.01** | 126.99 | 0.450 | 7.97 | 48.55 | 0.306 | 0.241 |
|   |   | ART-FR | 43.88 | 2.72 | 23.86 | **64.54** | **0.345** | **4.69** | **74.74** | **0.663** | **0.537** |
|   | 150 | DnCNN | 46.43 | 3.07 | **25.49** | 145.84 | 0.478 | 7.99 | 44.30 | 0.285 | 0.208 |
|   |   | ART-FR | **45.97** | **2.88** | 23.60 | **64.62** | **0.353** | **4.72** | **74.98** | **0.664** | **0.544** |
|   | 175 | DnCNN | 52.06 | 3.73 | **23.87** | 174.35 | 0.508 | 8.03 | 39.36 | 0.269 | 0.177 |
|   |   | ART-FR | **46.80** | **3.01** | 23.01 | **67.18** | **0.365** | **4.62** | **75.33** | **0.675** | **0.555** |
|   | 200 | DnCNN | 57.01 | 4.03 | **22.45** | 221.47 | 0.534 | 7.91 | 34.97 | 0.254 | 0.155 |
|   |   | ART-FR | **48.34** | **3.31** | 22.09 | **66.66** | **0.378** | **4.67** | **75.87** | **0.684** | **0.556** |

Figure 8: A **qualitative** comparison between ART-FR and DnCNN was conducted under severe degradation conditions that deviate from the training domain.

## C  MORE VISUAL RESULTS

### C.1  INPAINTING, COLORIZATION, AND OLD PHOTO RESTORATION

ART-FR can be flexibly applied to a wide range of low-level vision tasks, including image inpainting, colorization, and super-resolution. In this section, we present additional visual results in Fig. 9.

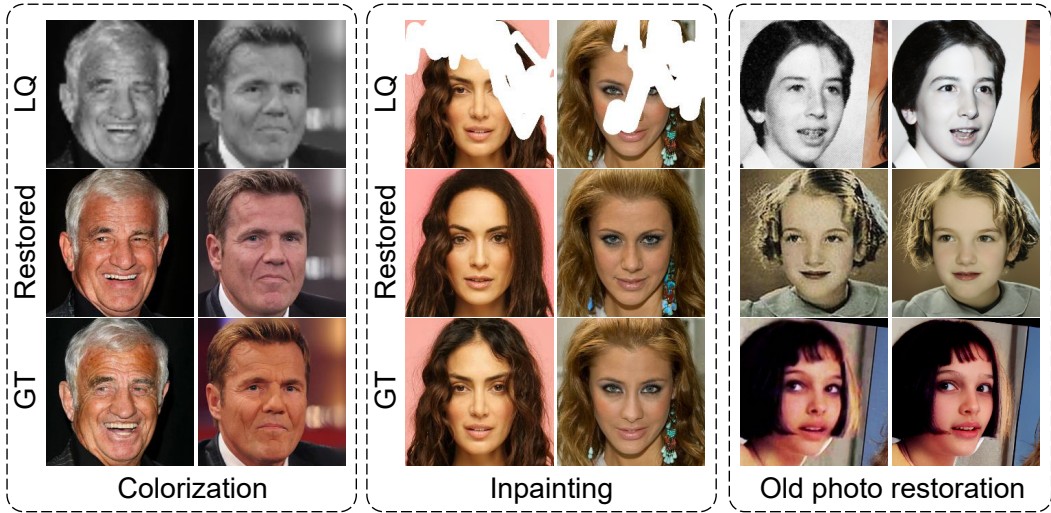

Figure 9: Visual results on the extended applications of ART-FR.

### C.2  ABLATION STUDY

Visual comparisons from the ablation study (Sec. 4.3) are presented in this section (Fig. 10).

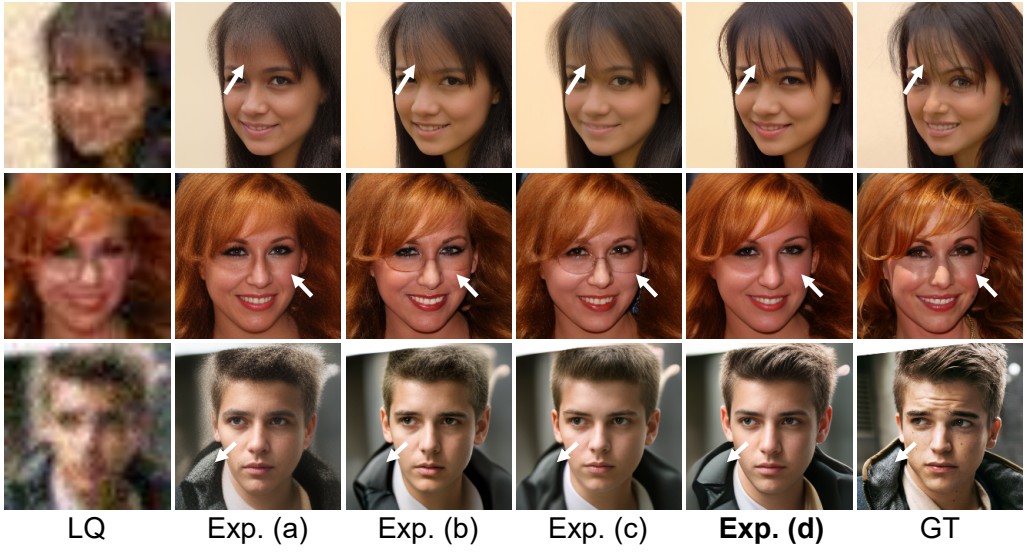

Figure 10: Visual comparisons of the ablation study.

### C.3  SYNTHETIC AND REAL WORLD DATASETS

More qualitative comparisons between ART-FR and previous works are provided in Fig. 11, 12.

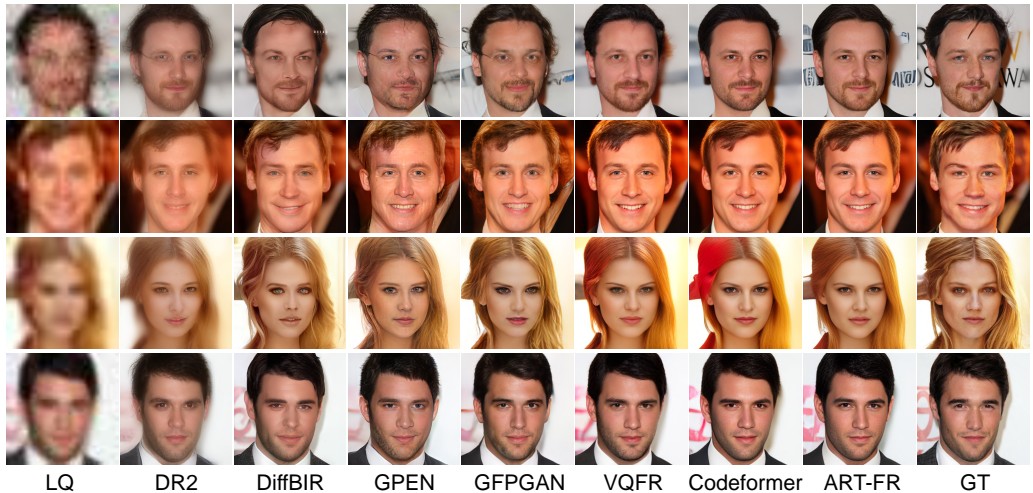

Figure 11: More qualitative comparison on the Celeb-Test.

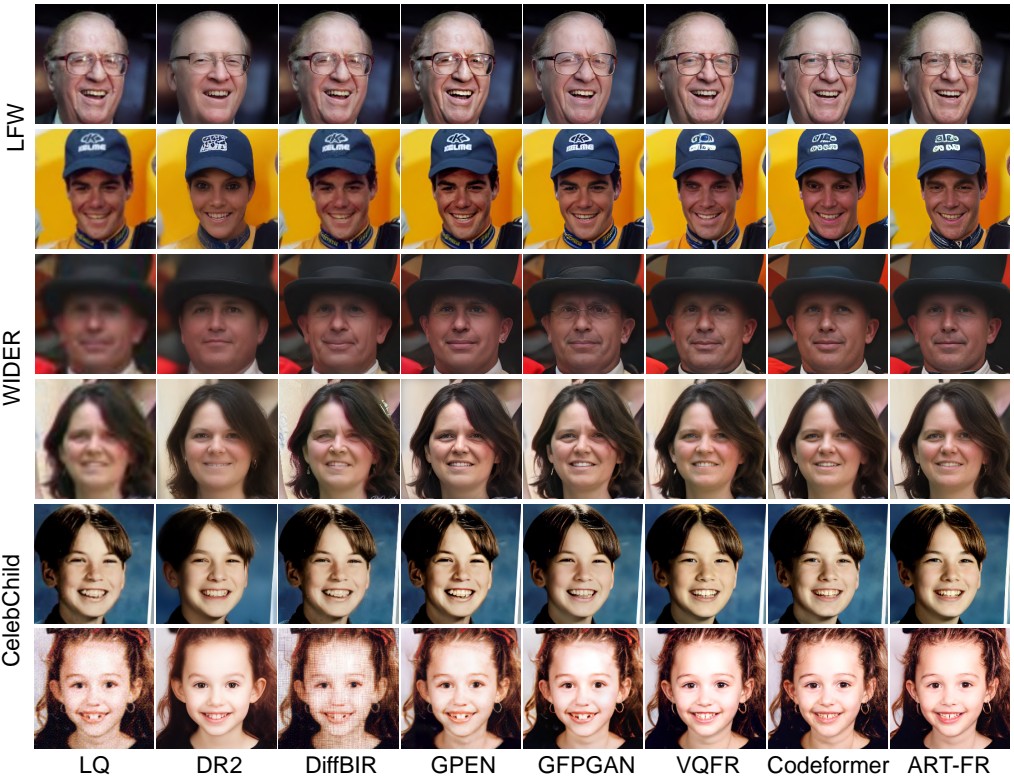

Figure 12: More qualitative comparison on real-world datasets.

# D  EXTENTIONS

## D.1  INFERENCE TIME

One of the advantages of the Masked Auto-regressive model in generative models is its fast inference speed. We compare the running time of the methods, as mentioned in the main text, with our proposed ART-FR. We evaluate these methods on $512 \times 512$ face images, and the results are shown in Table 6. ART-FR is significantly faster than diffusion-based methods but slightly slower than GAN-based and other VQ-based methods.

Table 6: **Inference time of different methods.** We evaluate these methods on an NVIDIA GeForce RTX 3090 GPU. The results show the average time required for these methods to restore an image.

| Methods | DR2 | DiffBIR | GPEN | GFPGAN | Codeformer | VQFR | **ART-FR** |
|---|---|---|---|---|---|---|---|
| Time (sec) | 2.3549 | 9.7055 | 0.0221 | 0.0226 | 0.0321 | 0.0748 | 0.2124 |

## D.2   SAMPLING DIVERSITY OF ART-FR

Sampling diversity is a critical evaluation metric for generative models. In auto-regressive models, a trade-off between sampling diversity and sampling quality can be achieved by introducing varying levels of sampling temperature into the sampling process, a method that is naturally integrated into ART-FR. As illustrated in Fig. 13, setting the sampling temperature for the model at 0, 1, 2, and 4 during the reconstruction process yields different outcomes. Through this approach, ARFR can achieve a compromise between fidelity and authenticity in reconstruction.

**Adding temperature**

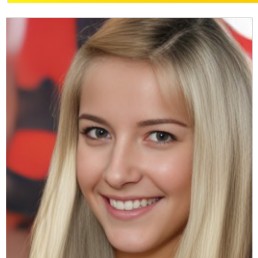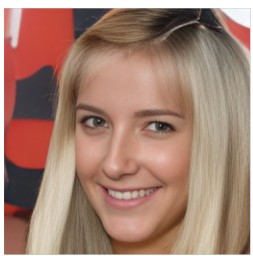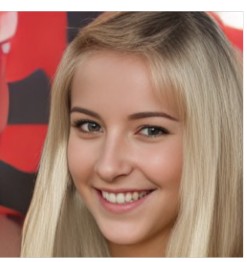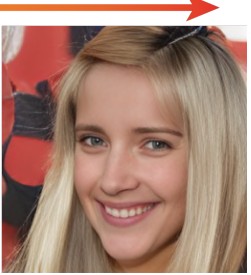

Figure 13: **ART-FR reconstruction results vary with changes in the sampling temperature.** By incrementally increasing the sampling temperature during the reconstruction process, ART-FR exhibits a wide range of diverse reconstruction outcomes.

## D.3   INCORPORATION OF LQ CONDITION

As the first approach to introduce auto-regressive transformer to image restoration, this work explored methods of incorporating LQ condition into auto-regressive models. Specifically, as shown in Fig. 14, the first method utilizes cross-attention module to inject the projected LQ feature into the network, guiding the image towards it. The second method concatenate the projected LQ feature, with HQ tokens that have been masked on the channel dimension. The third method connect LQ feature with masked HQ tokens to form a longer token sequence. These two sets of token sequences then continuously interact within Bidirectional Transformer module and are ultimately mapped to the predicted logits corresponding to HQ tokens. The specific comparison of these mothods is presented in Table 3. The results indicate that ART-FR is applicable to a variety of different LQ guidance methods. Ultimately, ART-FR adopted the first method which demonstrated more stable convergence during training.

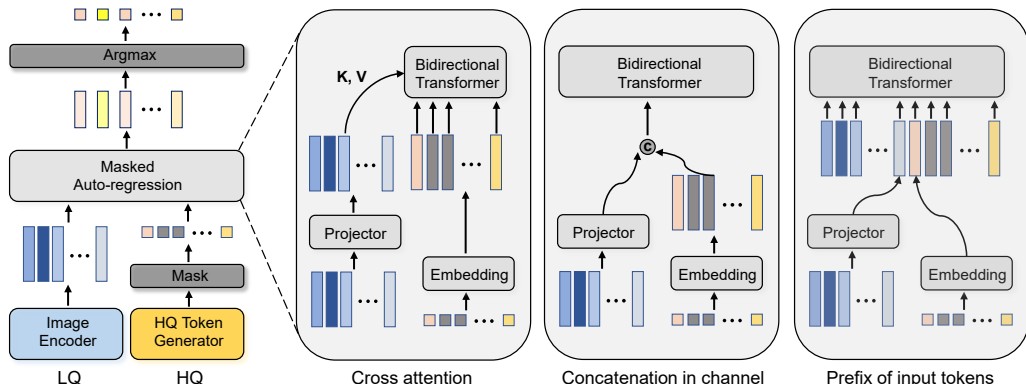

Figure 14: **Methods for incorporating LQ condition.** The first method involves inputting LQ image features as prefix tokens together with masked HQ tokens into the auto-regressive model; the second approach concatenate image features with tokens on the channel dimension; the third method employs a cross-attention module to introduce conditional information.

### D.4 LIMITATIONS AND FUTURE WORK

While ART-FR achieved promising results in face restoration, there are two scenarios where it may produce suboptimal outcomes. The first occurs when the degraded image includes hand features, which we attribute to the limited reconstruction capabilities of VQGAN. As illustrated in Figure 15 (a), directly reconstructing HQ images with VQGAN still leads to unsatisfactory results. Since our model relies on direct supervision from these reconstructed HQ images, enhancing the expressiveness of VQGAN represents a potential avenue for improving restoration performance.

The second scenario emerges when the degraded image includes details unrelated to facial features, as demonstrated in Figure 15 (b). ART-FR struggles to restore the logo on the hat, whereas DiffBIR, despite producing less accurate facial details, reconstructs the hat more effectively. This difference can be attributed to DiffBIR's reliance on a pre-trained Stable Diffusion model, which has been exposed to a vast array of images during training. As a result, refining the training datasets specifically for face restoration emerges as another potential direction for improving performance.

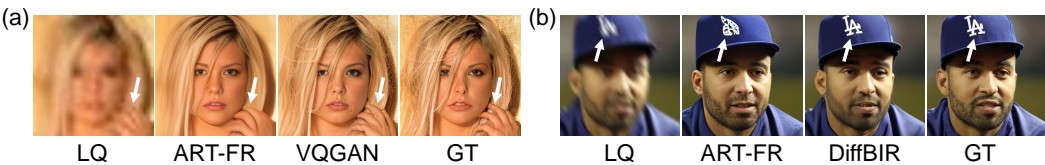

(a) LQ — ART-FR — VQGAN — GT   (b) LQ — ART-FR — DiffBIR — GT

Figure 15: **Failure outcomes of ART-FR.** ART-FR is constrained by the expressiveness of VQGAN and the incompleteness of the training datasets.

## E PSEUDOCODE FOR ART-FR

To provide a clearer understanding of the training process, we present the pseudo-code for ART-FR here.

---

**Algorithm 1** ART-FR

---

**Input:** HQ image $I_h$, LQ image $I_l$
1: **Train** LQ encoder $E_l(\cdot)$, masked auto-regressive model $A(\cdot)$
2: **Freeze** Quantizer $Q(\cdot)$, HQ encoder $E_h(\cdot)$, HQ decoder $D_h(\cdot)$
3: **while** not converged **do**
4:   **Generate gt information:**
5:     HQ feature $f_h = E_h(I_h)$
6:     HQ token sequence $q = Q(f_h)$
7:   **Forward:**
8:     LQ feature $f_l = E_l(I_l)$
9:     Masked HQ token sequence $q_M = M(q)$
10:    Predicted HQ token logits $p = A(q_M | f_l)$
11:  **Compute loss:**
12:    Feature loss $\mathcal{L}_{feat} = ||f_l - f_h)||_2^2$
13:    Cross-entropy loss $\mathcal{L}_{CE} = Cross - entropy(p, q)$
14:    Total loss $\mathcal{L}_{total} = \mathcal{L}_{feat} + \mathcal{L}_{CE}$
15:  **Backward:**
16:    $\mathcal{L}_{total}.backward()$
17:    optimizer.step()
18: **end while**

---

