# OpenReview forum: "ART-FR: masked Auto-Regressive Transformer for Face Restoration"
_ICLR.cc/2025/Conference — ICLR 2025 Conference Withdrawn Submission_

### Official Review · Reviewer_gGSo · 2024-11-01

**Soundness:** 3
**Presentation:** 3
**Contribution:** 3
**Rating:** 6
**Confidence:** 4

**Summary:**

The paper introduces ART-FR, a face restoration method that uses a Masked Auto-Regressive Transformer to address the challenging task of reconstructing high-quality facial images from degraded face images. The approach reformulates the restoration process as a conditional generation task in discrete latent space, enabling a more accurate mapping between low-quality and high-quality image representations. The model's structure includes a two-stage training process with a VQGAN-based visual tokenizer and a Masked Auto-Regressive model to progressively refine the latent representations. Experimental results show that ART-FR achieves superior performance across various degradation levels on both synthetic and real-world datasets, outperforming existing GAN- and diffusion-based approaches.

**Strengths:**

+ The paper proposes a unique approach by combining a visual tokenizer and auto-regressive transformer, which effectively reformulates face restoration as a generative task in discrete latent space. This framework is inspired by language models and reformulated to image restoration tasks.
+ ART-FR shows great capability to various levels of degradation, achieving high performance even under extreme noise conditions, as highlighted in the ablation and robustness studies.
+ Qualitative and quantitative results show ART-FR’s ability to preserve facial details accurately. ART-FR exhibits competitive scores across perceptual metrics, such as LPIPS and FID, and consistently outperforms other state-of-the-art methods in identity preservation.

**Weaknesses:**

+ Although ART-FR offers high performance, the auto-regressive process might incur significant computational overhead, particularly during inference, which may limit its deployment on devices with limited resources.
+ The paper does not fully address the potential for overfitting or instability in the generative model, especially in cases where the degradation in the training dataset does not match real-world data.
+ The comparison with DnCNN seems not reasonable, as DnCNN has lightweight parameters and is not designed for face restoration tasks. Maybe the authors should use other works to show the robustness of this work.

**Questions:**

- Although the experiments show superior performance of this method, the motivation for using auto-regressive in face restoration tasks seems not clear. Why did the auto-regressive manner benefit the SR tasks?
- Are there any failure cases if the degradation is light or severe? The authors give two scenarios in the appendix. But these are caused by the occlusion of hand and non-face regions. I wonder whether the auto-regressive manner could generate realistic facial images no matter whether the LR image is degraded.
- There seem few analyses about parameters T and N. Will they have an effect on the SR results?

---

### Official Review · Reviewer_sCLK · 2024-11-03

**Soundness:** 3
**Presentation:** 3
**Contribution:** 2
**Rating:** 5
**Confidence:** 3

**Summary:**

This paper reformulates the image restoration problem as a conditional generation problem and establishes a conditional auto-regressive modeling framework for face restoration tasks. The framework first obtains discrete representations of image by a VQGAN. And then, it uses an auto-regressive model to map low-quality image to high-quality image.

**Strengths:**

1. The paper proposes a new formulation of auto-regressive model for image restoration.
2. This method leads to consistent performance improvement across diverse face restoration tasks.

**Weaknesses:**

1. My major concern is the motivation to use an auto-regressive model for face restoration. The auto-regressive model is good at causal generation tasks. As for the issue of mapping problem for discrete latent space, I think it is related to transformer framework. The authors are suggested to provide more analysis on the motivation of applying an auto-regressive model for face restoration.
2. The experiments in the paper are not convincing. The main quantitative comparison in Table 1 is not convincing enough. The qualitative results in Figure 5 seem comparable with VQFG. Some hyper-parameters are suggested to construct more analysis (e.g., inference step $T$).
3. The inference speed of auto-regressive modeling method is not discussed, compared with other non-ar methods.

**Questions:**

1. As said in W1, how about the sequence-to-sequence mothed, which is also valid in discrete modeling problems?
2. In the inference, how does the inference step $T$ impact? And the impact of mask prediction order?
3. Some typos. Line 434, the quotation marks are reversed.

---

### Official Review · Reviewer_oZ2G · 2024-11-04

**Soundness:** 3
**Presentation:** 3
**Contribution:** 1
**Rating:** 3
**Confidence:** 5

**Summary:**

The paper presents an application of masked auto-regression transformer for VQGAN based face restoration. It is a topic of interest to the researchers in the related areas but the paper needs very significant improvement before acceptance for publication.

**Strengths:**

1. a new AR model for face restoration
2. competing performance on face restoration

**Weaknesses:**

1. This method is a direct improvement to the transformer module in CodeFormer method with limited contribution in methodological research. Also, the author may need to tell the difference between this method with the one in CVPR2023 paper "Not All Image Regions Matter:Masked Vector Quantization for Autoregressive Image Generation".
2. The writing structure and context is similar with Codeformer, and the material was not properly organized. For example, in Figure 3, the framework diagram seems good but the presented logic is confusing. The whole working process and training pipeline is not clear enough.
3. The presentation exists some misleading arguments. For motivation the author argues that "these methods struggle to accurately capture the mapping between low-quality(LQ) and high-quality(HQ) images in the discrete latent space, leading to suboptimal results". However, the measurement of this "accuracy" is not given and the mechanism of how domain gap influences latent representation is not stated.
4. For the above reason, the presentation should be focused on the highlights of results. Unfortunately, the presentation is far from acceptable for publication. LPIPS included in realness metrics is not reasonable and the experiment results of some listed methods seems abnormal.
5. The comparison methods are only until 2022, please compare with RestoreFormer(2023), DAEFR(2024), DifFace(2024) or more didn't show overall improvements for all metrics and the selection bias is not stated.
6. The ablation study is not enough comprehensive, lacking the comparsion with the model taking in Ground Truth token as inputs.

**Questions:**

see the weaknesses

---

### Official Review · Reviewer_wDYD · 2024-11-04

**Soundness:** 3
**Presentation:** 3
**Contribution:** 2
**Rating:** 3
**Confidence:** 4

**Summary:**

In this article, the authors propose an Auto-Regressive Transformer based Face Restoration (ART-FR) method, which reformulates the face restoration problem as a conditional generative task within a discrete latent space.

**Strengths:**

1.The paper is clearly written, and the method is detailed and technically sound.
2.The comparison with existing methods is comprehensive. There are also more visual results attached in the supplementary file.

**Weaknesses:**

1.Need to pay more attention to in the layout of the article, including the lack of units for PSNR and misaligned fonts in the tables (Table2 & Table3).
2.The method used in the article is not innovative enough and the visual effect is not satisfactory. Taking Figure 4 as an example, the comparisons are not obvious.
3.The Ablation Study section of the article is not rigorous enough and lacks experimental numerical controls. The author should provide the ablation results for each module.

**Questions:**

No more questions.

---

### Note · Authors · 2024-11-13

I have read and agree with the venue's withdrawal policy on behalf of myself and my co-authors.